# A Bibliometric Analysis of Augmented Reality in Language Learning

**Wenhe Min and Zhonggen Yu \***

Faculty of Foreign Studies, Beijing Language and Culture University, Beijing 100083, China;
202221198373@stu.blcu.edu.cn
\* Correspondence: yuzhonggen@blcu.edu.cn

**Abstract:** As an emerging technology, augmented reality has been increasingly entering the educational field. Previous studies of AR are strongly associated with scientific education but lack the mention of language learning. This study aimed to analyze the use of AR tools in language learning contexts using the bibliometric tools VOSviewer and CitNetExplorer using both qualitative and quantitative research methods. This study analyzed the top ten authors, sources, countries, and organizations using VOSviewer and established citation networks using CitNetExplorer. The study found that both teachers and students showed a positive attitude toward AR tools in language learning. Games, 3D images, and videos are the main ways to instantiate virtual elements into the real world, and the most widely used tool in AR-assisted learning was HP Reveal. Moreover, the study also found that AR tools could enhance language learning by presenting an immersive learning context, increasing motivation, providing interaction, and reducing anxiety. Future research will contribute to how AR tools influence students' learning performance and teachers' teaching activities.

**Keywords:** bibliometric analysis; augmented reality; VOSviewer; CitNetExplorer; language learning

## 1. Introduction

With new educational technologies emerging, more opportunities are offered for computer-enhanced language learning approaches [1]. Research on AR in language learning began in 2008 and continues to progress. As a newly introduced technology, AR plays an important role for students in language learning. According to [2], one of the barriers of learning English in non-English speaking countries is the lack of real contexts outside of the classroom. For this problem, AR can instantiate virtual elements into the real world and create a real English-learning context for learners. Furthermore, statistical analyses show that teaching with AR technology can help students improve their motivation in language learning, which can directly influence their performance [3]. Aside from the benefits for students, AR can also offer educators new opportunities to engage learners in innovative ways [4]. Thus, it is important to develop a framework for conceptualizing and promoting the use of AR in language learning.

As an emerging technology, augmented reality is increasingly entering the field of education. However, the language skills currently used in AR technology require relatively lower cognitive engagement, such as pronunciation, meaning comprehension, and word recognition. There is a lack of more complex language skills used in the AR applications, such as reading and writing [5]. Additionally, according to the search results from Web of Science, there are much more meta-analyses and systematic reviews than bibliometric studies (shown in Table 1). Table 2 shows that the research conducted by [6] focused on the performance and production of AR in the field of education, with 777 publications involved. Another research conducted by [7] aimed to find out the importance of English mobile learning. A total of 5434 publications were analyzed. However, these research studies only introduced the importance and production of AR tools without exploring the actual use

of AR in language learning. Therefore, a study combining a bibliometric analysis and the application of AR with a sufficient number of included publications is needed.

**Table 1.** The number of publications of different review studies on AR tools in language learning.

| Study Type | Time Period | Number of Publications |
|---|---|---|
| Meta-analysis | 2013–2022 | 30 |
| Systematic review | 2016–2022 | 48 |
| Bibliometric analysis | 2020–2022 | 2 |

**Table 2.** The comparison between previous bibliometric studies on AR-enhanced language learning and this study.

| Authors/Year | Highlights | Research Topics | Number of the Publications Included | Bibliometric Tools |
|---|---|---|---|---|
| [6] | The performance and production of AR in the field of education | Top 5 knowledge fields; document type; institutions; authors; sources; countries and most cited articles on AR in the field of education; and the way to teach AR effectively. | 777 | Science Mapping Analysis Tool (SciMAT) |
| [7] | The importance of English mobile learning | Research direction and type; years; countries; productive institutions, top authors, and co-occurrence; co-authorship, citation, and co-citation; top 10 cited sources; and the top 20 keywords through VOSviewer. | 5343 | VOSviewer and CiteSpace |
| This study | The use of AR tools in language learning | Annual trend of publications; top 10 authors, sources, organizations, and countries; users' attitudes; ways to integrate with language teaching and learning; and the effects of AR. | 1275 | VOSviewer and CiteSpace |

The bibliometric analysis was performed using the application of quantitative techniques for domain-specific bibliometric data. Software such as VOSviewer and CitNet-Explorer are essential for bibliometric analyses. These programs provide bibliometric graphs that show co-authorship, citations, and bibliographic links between papers by analyzing data downloaded from digital databases, which can be visualized to quickly focus on hot topics in a particular field. This method has many advantages in terms of efficiency and accuracy over traditional quantitative methods. It allows for a comprehensive and objective identification of research gaps and hot issues in a given field while reducing the cognitive load on the reader.

The analysis of this study combines quantitative and qualitative research methods [8]. At the beginning of this study, the use of AR in e-learning was analyzed through VOSviewer, including the trend of publications and citations, as well as the top ten authors, sources, organizations, and countries. Secondly, citation networks were created by using clustering techniques; the two longest citation paths were drilled down using CitNetExplorer. Finally, according to the two longest citation paths, conducted a bibliometric analysis to examine users' attitudes toward AR, the effects of AR use, and how language learning is integrated with AR.

The current study is composed of six parts. Section 1 is the introduction of the study, which includes the background information, significance of the study, two bibliometric tools of the study, and overall structure of the study. Section 2 provides a general statement about the literature review, which includes the previous studies of AR, AR in language learning, and users' attitudes toward the technology; the research purpose and questions are also proposed in this chapter. Section 3 contains the research methodology; it consists of the scope of the study, application of the two bibliometric tools, and procedure of data collection and data analysis. Section 4 shows the results of the bibliometric analysis, which is followed by the discussion of the research results in Section 5. Section 6 provides the

conclusion of the study and includes three parts: major findings, limitations of this study, and implications for future research.

## 2. Literature Review

### 2.1. Previous Studies of AR

The studies of AR can originate back to the 1950s. The term AR was introduced by Tom Caudell in 1990. Subsequently, Ref [9] found the reality–virtuality (RV) continuum, which combines reality and virtuality. In the RV continuum, the real environment is situated on the left side, which solely includes the environment consisting of real objects. The virtual environment is on the right side of the RV continuum, which solely consists of virtual objects. In this framework, mixed reality (MR) is defined as anywhere on the RV continuum between these two extreme environments. Within the class of MR, augmented reality is analogous but antithetic to augmented virtuality. Therefore, under the background of the RV continuum, it would be more clear to understand the definition of AR such that it is "augmenting natural feedback to the operator with simulated cues" [9]. In 1997, Azuma conducted the first study on the use of AR. In recent years, AR has developed at a high speed with the emergence of mobile applications [10]. AR produces the best of the natural environment and available virtual information. The elements of reality and virtuality in AR offer users a new world supported by virtual information.

AR is an enhanced, interactive version of a real-world environment, which is achieved by using digital elements; this means that AR can help users see virtual objects as composited within the real environment [11]. In fact, AR technology is not limited to a specific type, nor is it restricted to vision; it can also be used to enhance smelling, touching, and hearing [12]. Nowadays, AR has been applied in many fields in modern society, including advertising and marketing [13], architecture [14], and entertainment [15]. It has also been adopted in education [16,17], such as in the sciences, in math, and in literacy [18,19]. Augmented reality is applied by learners at different stages of schooling, ranging from primary school to college students [20]. Although numerous research studies have been conducted to investigate the importance and effect of AR, there is a lack of a bibliometric analysis of the findings with respect to language learning. With the development of research in this field, researchers are working hard to discover the issues and trends across the entire field. Therefore, it is urgent and necessary to develop a bibliometric study to provide directions for future study.

### 2.2. AR in Language Learning

AR is a real-time representation using computer-generated images, sounds, or videos that allows users to interact with the real world in an augmented way [10]. As an assistant tool, the application of this technology is very important in language teaching and learning [21] because mobile applications for touch screens can offer children new opportunities for language learning [22]. However, foreign language teaching is difficult and needs effective skills to avoid boredom. AR can provide an augmented learning context, which can help children enrich their language learning experiences through the combination of virtual and real environments. Ref [23] conducted a study on collaborative modeling in augmented reality. The authors found that the use of AR tools may have a positive influence on learners' perceived efficacy and self-perception, which can directly impact learners' performance. The authors also pointed out the limitations of AR tools in that the environment within AR may depend on students' background knowledge and their familiarity with real world settings.

However, the above studies have commonly focused on the benefits and limitations of AR applications while neglecting how these technologies can help children and how the AR technologies were integrated into the teaching and learning. Based on the application of AR in previous studies, this paper will systematically explore the effect of AR on language learning and the ways to integrate AR tools with language learning.

### 2.3. Users' Attitude

Research has shown that users' attitudes toward educational technology has a great effect on technology adoption [24]. This is because teachers' negative attitudes affect their way of thinking and directly hinder the integration of technology in language teaching. In contrast, teachers with positive attitudes toward the use of a technology are more likely to adopt it. Ref [24] found that the same applies to learners. Students who use technology to learn can show higher levels of motivation and interest. Previous studies have found that most teachers and learners behave positively toward AR tools in the language classroom [11]; however, this previous research only included studies published before 2019. Therefore, this paper will examine users' attitudes toward AR use in language learning based on recent publications.

### 2.4. Research Purpose and Questions

According to the literature review, this study will analyze the AR use in language learning from a bibliometric point of view and explore the users' attitude toward AR, ways of integrating AR, and effectiveness of AR. The research questions are as follows:

RQ1: What is the year-based trend of included publications and citations?

RQ2: What are the top ten authors, sources, organizations, and countries among the studies on AR in language learning?

RQ3: What are the users' attitudes toward AR tools in language learning?

RQ4: What are the effects of AR on language learning?

RQ5: How are AR tools integrated with language learning?

## 3. Methods

### 3.1. Defining the Scope

Defining the purpose and scope in the bibliometric analysis is the first thing that needs to be done [25]. In this study, authors have retrospected the research achievements and science in this field, which means unpacking the rich research constituents and uncovering the networks between the constituents in each research. Moreover, according to [25], the scope of a bibliometric analysis needs to include 500 papers or more. Therefore, the number of analyzed publications in this study will be over 500.

### 3.2. Technical Tools

In this study, we attempted to bibliometrically analyze the use of AR by using VOSviewer and CitNetExplorer. We analyzed the citation networks and performed a bibliometric analysis with the help of CitNetExplorer and VOSviewer [26]. CitNetExplorer focuses on analyzing and visualizing a citation network, which can show the publication year, citation score, publication group, cluster publications based on citation relationships, and can drill down the large citation network; meanwhile, VOSviewer mainly focuses on visualizing bibliometric networks based on citation, co-citation, co-occurrence, and co-authorship. Both methods allow for a visualization of the clustering results. The assignments of publications into different clusters are linked to the relatedness of publications. In this study our clustering technique CitNetExplorer does not take the direction of a citation into account, which means that there is no distinction between publication i citing publication j and publication j citing publication i [8]. The clustering parameters in CitNetExplorer contain a resolution parameter, which is 1.00, and a minimize cluster size parameter, which is 10. Moreover, the optimization parameters include: the number of random starts, which is 1; the number of iterations, which is 10; and random seed, which is 0. Due to the defined minimum size, 94 publications were not included in the clustering. CitNetExplorer tends to cluster at the individual level, while VOSviewer mainly clusters at the global level of bibliometric analysis [27]. Both methods allow for a visualization of the clustering results.

### 3.3. Data Collection

Instead of key words, citation networks were used to identify the relationship between publications. Because of the difficulty of clarifying the exact meaning, keywords were not used to define relationships. Keywords can be interpreted differently in different contexts. This can lead to ambiguity in the classification of clusters, causing difficulties in the analysis of clustering decisions. Because co-citation and bibliographic coupling are the indirect estimates of relationships, they do not offer direct and precise information. Therefore, they cannot be used as criteria for determining relationships between publications [28]. CitNetExplorer clusters publications according to their relationships. To avoid technical or analytical problems, we assigned some publications in this study to unique groups [8].

The procedure of data collection was as follows. On 13 November 2022, we searched the Web of Science tool by entering "AR" OR "augmented reality" (topics) and "language learn*", "language teach*", and "education", obtaining 1318 open access results, which were used in the bibliometric analysis with the help of VOSviewer. The document types included articles (N = 1101), review articles (N = 232), early access articles (N = 43), editorial materials (N = 24), proceeding papers (N = 11), book chapters (N = 6), corrections (N = 2), letters (N = 2), book reviews (N = 1) and meetings (N = 1). A total of 1318 articles were found using the Web of Science tool. Studies were included if they (1) focused on AR and language learning, (2) provided sufficient information for the study, (3) were written in English, and (4) contained convincing results. Studies were excluded if they were (1) duplicates, (2) irrelevant, (3) not relevant to the research question, (4) written in another language, (5) not full texts, or (6) not relevant to the education sector. Based on these criteria, 1318 titles and abstracts were evaluated. Finally, 1275 articles were chosen for further bibliometric analysis using CitNetExplorer.

The timeline ranged from the inception of the online databases to the year 2022. The researchers developed the search strategies and obtained literature by searching the Web of Science tool on 13 November 2022. Web of Science includes many databases, such as the Science Citation Index Expanded, Social Sciences Citation Index, Arts & Humanities Citation Index, Conference Proceedings Citation Index-Science, Conference Proceedings Citation Index-Social Science & Humanities, Emerging Sources Citation Index, Current Chemical Reactions, and Index Chemicus. Therefore, the use of Web of Science could reduce the selection bias and improve the representativeness of the included studies [24].

To increase reliability, the publications that were included were chosen from a variety of sources and were available in different languages. The main sources included *Computers & education* (N = 221), *Educational Research Review* (N = 134), *Educational Technology & Society* (N = 89), *Education Science* (N = 68), *Interactive Learning Environments* (N = 64), *Computers in Human Behavior* (N = 50), *Personal and Ubiquitous Computing* (N = 50), *International Journal of Environmental Research and Public Health* (N = 31), *Eurasia Journal of Mathematics, Science and Technology Education* (N = 26) and *Clinical Anatomy* (N = 21). The languages of included documents included English (N = 1265), Spanish (N = 72), Portuguese (N = 6), Russian (N = 6), Turkish (N = 6), Ukrainian (N = 3), German (N = 3), Chinese (N = 2), Hungarian (N = 1), and Slovak (N = 1).

### 3.4. Procedure of Data Analysis

The procedure of the content analysis was as follows. First, we made use of the citation report in the Web of Science tool, and the times cited and publications over time were provided. Based on the citation report, we analyzed the publication trend of the search topics. Second, 1275 selected articles were inputted into VOSviewer. Later, we chose the co-occurrence analysis to bibliometrically analyze the top ten authors, sources, organizations, and countries. According to the co-occurrence result, we analyzed the situation of AR learning development. Third, we imported the 1275 articles into CitNetExplorer to bibliometrically analyze the citation network. The author's information, title, and source could be found in CitNetExplorer. Then, we chose two publications to drill down to deter-

mine the longest path. In this way, we found out the common focus of the publications in the longest path. Based on the focus, we explored the answers in the publications.

## 4. Results

RQ1: What is the year-based trend of included publications and citations?

The survey results showed the number of annual publications, which were used to analyze trends in the included and cited articles. The survey results included 123 review articles and 394 open access articles covering the period ranging from 2008 to 2022. Figure 1 shows that research on AR in language learning has been growing since 2008, when research on AR in language learning was born. The number of relevant studies fluctuated between 2008 and 2010, but suddenly increased in 2011. From 2011 to 2014, the number of relevant studies also fluctuated up and down. From 2015 to 2021, the number increased steadily and reached a peak in 2022. Since then, the number of relevant research has gradually increased. Between 2019 and 2022, there is a large number of publications on the topic of AR in language learning. From the birth year of AR until 2022, the number of citations continues to increase, especially in 2020 and 2021. Therefore, according to the publication trend, the topic of the use of AR in language learning will continuously be a hot issue. Future researchers can target their research directions based on the figure below.

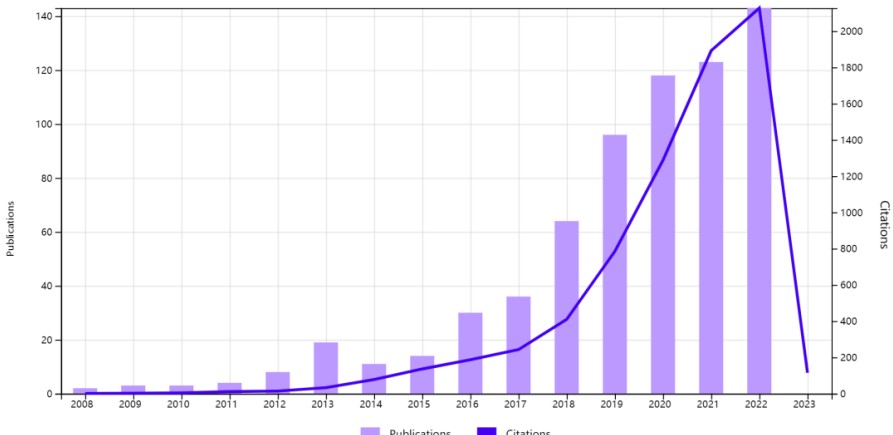

**Figure 1.** Times cited and publications of search topics over time.

RQ2: What are the top ten authors, sources, organizations, and countries among the studies on AR in language learning?

To fill the research gaps in each country, we identified the most cited authors, sources, organizations, and countries in this field using VOSviewer (see Table 3). In addition, we clustered the citation strength of the most cited authors, sources, organizations, and countries that contribute most to the field. It is important to note that researchers can find a large number of high-cited published citations from these highly cited sources. The cluster analysis provides a comprehensive overview of the most cited studies.

As Table 3 shows, the top ten cited authors are Chang Hsin-Yi, Lee Silvia Wen-Yu, Liang Jyh-Chong, Wu Hsin-Kai, Akcayir Gokce, Akcayir Murat, Baldiris Silvia, Kinshuk, Bacca Jorge, and Fabregat Ramon. The top ten cited sources are *Computer & Education*, *Educational Research Review*, *Educational Technology & Society*, *Computers in Human Behavior*, *Personal and Ubiquitous Computing*, *Eurasia Journal of Mathematics, Science and Technology Education*, *International Journal of Environmental Research and Public Health*, *Interactive Learning Environments*, *Education Science*, and *Clinical Anatomy*. The top ten organizations are National Taiwan University of Science & Technology, National Taiwan Normal University, National Changhua University of Education, National Kaohsiung Normal University, Kirikkale University, Gazi University, University of Girona, Athabasca University, University of La Laguna, and University of Granada. The top ten countries are the USA, Spain,

Taiwan, Turkey, the People's Republic of China, Canada, Australia, South Korea, England, and Greece.

**Table 3.** Top ten authors, sources, organizations, and countries.

| N | Cited Author | Citation | Link | Cited Source | Citation | Link |
|---|---|---|---|---|---|---|
| 1 | Chang, Hsin-Yi | 860 | 316 | *Computers & Education* | 1841 | 221 |
| 2 | Lee, Silvia Wen-Yu | 860 | 316 | *Educational Research Review* | 636 | 134 |
| 3 | Liang, Jyh-Chong | 860 | 316 | *Educational Technology & Society* | 480 | 89 |
| 4 | Wu, Hsin-Kai | 860 | 316 | *Computers in Human Behavior* | 315 | 50 |
| 5 | Akcayir, Gokce | 577 | 351 | *Personal and Ubiquitous Computing* | 302 | 50 |
| 6 | Akcayir, Murat | 577 | 351 | *Eurasia Journal of Mathematics, Science and Technology Education* | 288 | 26 |
| 7 | Baldiris, Silvia | 565 | 378 | *International Journal of Environmental Research and Public Health* | 256 | 31 |
| 8 | Kinshuk | 565 | 378 | *Interactive Learning Environments* | 207 | 64 |
| 9 | Bacca, Jorge | 565 | 378 | *Education Science* | 160 | 68 |
| 10 | Fabregat, Ramon | 524 | 308 | *Clinical Anatomy* | 158 | 21 |
| N | Organization | Citation | Link | Country/Region | Citation | Link |
| 1 | National Taiwan University of Science & Technology | 919 | 180 | USA | 2042 | 332 |
| 2 | National Taiwan Normal University | 912 | 205 | Spain | 1877 | 397 |
| 3 | National Changhua University of Education | 873 | 184 | Taiwan | 1370 | 248 |
| 4 | National Kaohsiung Normal University | 860 | 177 | Turkey | 1212 | 388 |
| 5 | Kirikkale University | 578 | 227 | People's Republic of China | 635 | 216 |
| 6 | Gazi University | 577 | 217 | Canada | 616 | 159 |
| 7 | University of Girona | 526 | 178 | Australia | 450 | 75 |
| 8 | Athabasca University | 508 | 165 | South Korea | 374 | 70 |
| 9 | University of La Laguna | 451 | 73 | England | 286 | 83 |
| 10 | University of Granada | 365 | 132 | Greece | 236 | 170 |

Five clusters were identified based on the clustering methods in CitNetExplorer. Publications were clustered based on citation links. After selecting the 100 most cited publications in the citation network visualization, Figure 2 was created, which contains 1275 publications (only 100 could be visualized) and 10,890 citation references. Due to the defined minimum size, 94 publications were not included in the clustering. Table 4 shows the number of publications in each group, number of citations, number of publications cited more than 20 times, and number of publications in the top 50 most cited (see Table 4). The citation network covers the period ranging from 1951 to 2022.

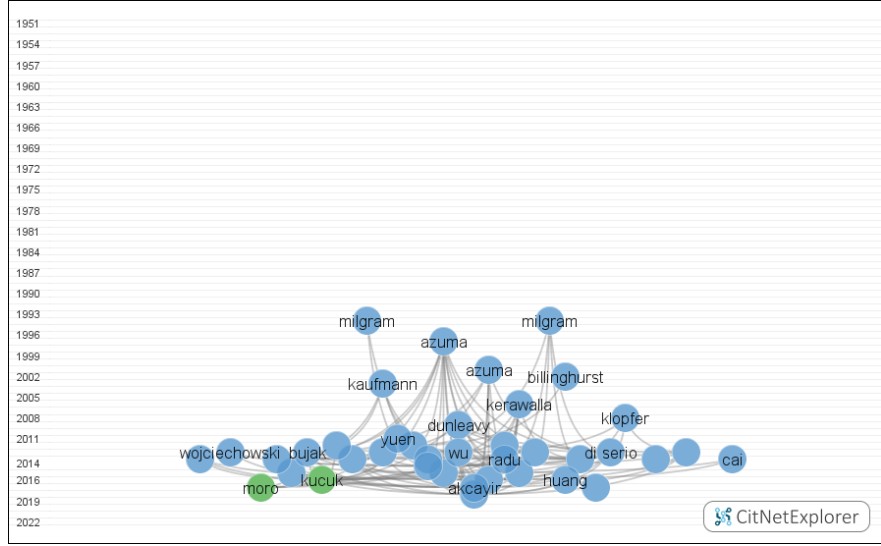

**Figure 2.** Visualization of citation networks.

**Table 4.** Five clusters and citation networks.

| Cluster | Color | No. of Publications | No. of Citation Links | No. of Publications $\geq$ 20 Citations | No. of Publications in 50 Most Cited Publications |
|---|---|---|---|---|---|
| 1 | Blue | 830 | 7193 | 104 | 40 |
| 2 | Green | 192 | 716 | 12 | 3 |
| 3 | Purple | 89 | 352 | 7 | 0 |
| 4 | Brown | 48 | 157 | 3 | 7 |
| 5 | Yellow | 14 | 17 | 0 | 0 |

RQ3: What are the users' attitudes toward AR tools?

The authors used CitNetExplorer to drill down the longest paths. It is generally accepted that the longest path between two articles indicates the citation relationship between them and reveals hot topics in the research area. After searching for the longest path, the authors identified the citation relationships of 11 publications in Figure 3. AR-enhanced education was first proposed as a teaching approach by [29]. According to [30], the co-citation network of cited authors can show a connected focus among these authors in a given field. The other publications mainly focus on the analysis of technologies, strategies, and methods of using AR tools, as well as the evaluation and attitude toward AR. Thus, it indicates that the integrated method of AR tools and attitude toward AR are popular topics in the research field.

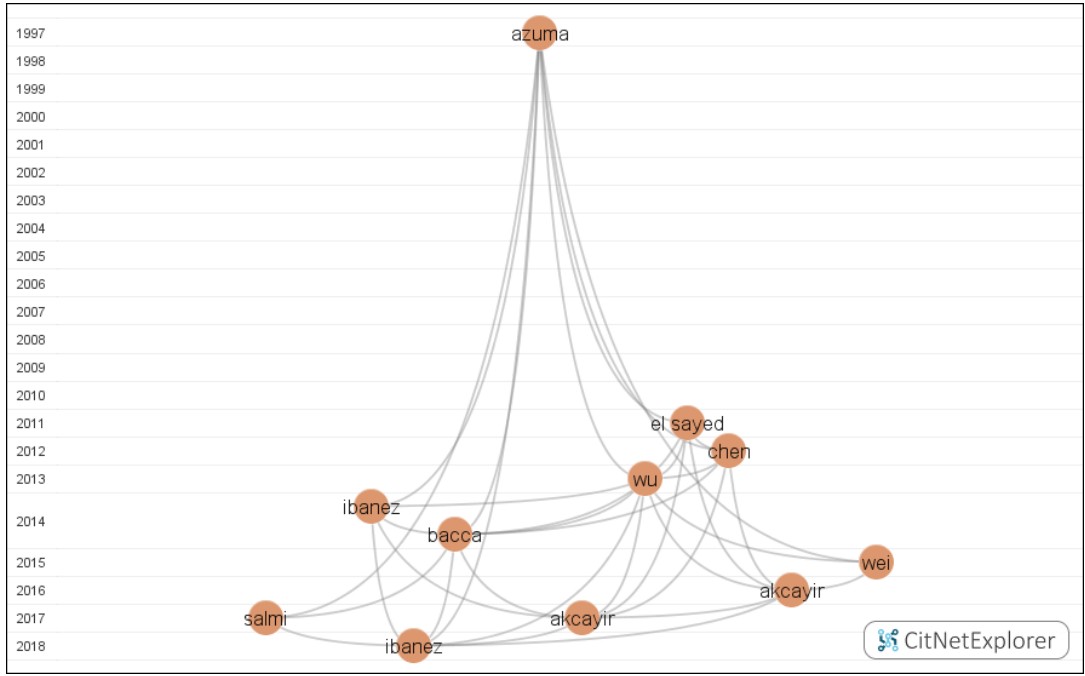

**Figure 3.** Visualization of the first longest path analysis.

AR technology combines elements of real and virtual content and has the potential to address the lack of real-life scenarios outside of the classroom. According to [31], attitude was the second most influenced emotional state in AR learning environments. The author also found that AR can foster students' positive attitude toward learning, which can promote their performance. Ref [32] pointed out that most students in their study hold a positive attitude toward the use of AR in their language learning; they thought it was enjoyable to learn through play. Ref [33] proposed that children often enjoy augmented reality media and that this has a positive impact on their learning. After experiencing AR tools in the classroom, students showed a clear tendency to prefer them as language learning tools and showed more interest in the learning process [34]. However, the use of

augmented reality in the classroom is not limited to the learner but instead largely depends on the teacher's willingness and ability to use it effectively [2].

Ref [35] noticed the important interaction between teachers' attitudes and intentions and found that a positive attitude toward AR-supported instruction influences its continued application in the classroom. They found that teachers' positive attitudes toward the application of AR included satisfaction, appropriateness, reliability, and belief. Positive attitudes toward AR technology predict future educators' digital confidence in implementing AR. In previous studies [36], the ratings of reliability in AR use were higher than the ratings of satisfaction and relevance. However, the new dimension "belief" was significantly better than the other three dimensions. This suggests that teachers are convinced of the need to integrate augmented reality into teaching, including in content creation and visualization in the coming years.

RQ4: How are AR tools integrated with language teaching and learning?

Except for the attitude toward the use of AR, the publications in the first longest path (Figure 3) also commonly discussed the ways to integrate AR with language teaching and learning. According to [37], the AR systems that are currently used are portable technologies, which can improve the immersion experience. As the authors mentioned, TimeLab2100, which provides real-world locations (e.g., a subway station and a park), can help students find a substitute for their learning needs. Ref [32] found that the well-integrated ways, including images, texts, and videos, could better prevent students' cognitive loads and improve their academic performance. Ref [38] proposed a new way to integrate AR tools with language learning. The authors designed an AR library institution system that combines 3D technology and a library environment to teach students skills. This innovative way can improve teachers' teaching effectiveness and make it attractive for students to learn.

Augmented reality tools are closely related to different aspects of language learning. In a study by [39], HP Reveal, a platform that delivers virtual content to users through mobile devices, was identified as the most widely used tool for computer-assisted learning. Teachers use this advanced platform to convert images into 3D and allow students to visualize learning material. In addition to HP Reveal, some researchers have used Vuforia, an augmented reality application, and Unity, a game engine used in language studies, to develop learning content. Teachers can use these two tools to create AR content for language learning. Ref [39] also mentioned that learner immersion should be the main focus of language learning in augmented reality research.

AR enhances language learning by providing learners with virtual elements that are superimposed upon the real environment in the way of 3D images, movies, and games. AR applications can provide 3D images with sound and movement and are often used to teach vocabulary and writing. In a study by [40], an AR app with 3D images was used to teach brushstroke spelling via animation. It helped students learn by explaining step-by-step how to write Chinese characters, and the tool developed their spelling acuity. In addition, AR videos have also been used to foster language learning. Ref [41] conducted a study using AR videos to support students' English language learning. When students studied language with the AR tool, an AR video of the corresponding objects appeared on the screen. The results showed that this helped students learn the learning material in a clearer and more concrete way, which significantly improved learning outcomes. Augmented reality games have also been integrated into language learning. According to [39], the most widely used games for language learning are Pokémon Go and ChronoOps. Pokémon Go and ChronoOps are location-based augmented reality mobile games. In the game scenarios, students can learn the language by looking at the names of three-dimensional objects and listening to their pronunciation.

RQ5: What are the effects of AR on language learning?

To answer this question, this study drilled down the publications and created another citation network according to [9] (see Figure 4). This citation network consisted of eight publications. After eliminating studies similar to the first longest path, we focused on those

by [31,37,42–44]. It was found that all eight of these publications discussed the effects of the use of AR tools. These articles mainly focus on students' motivation, interests, and performance. Ref [42] developed an AR-supported learning system called HELLO, which aims to improve students' language skills. The results showed that the AR-supported system can provide more learning resources for students and provide enjoyable learning experiences. Based on the results, it was concluded that the use of AR tools has an effect on language teaching and learning. Ref [37] also explored the learning effects of AR tools; the authors thought that, when compared with other technology-supported environments, an AR system can help learners to improve their language skills in a more effective way. It can show learners the learning content in a 3D method so that learners can manipulate the virtual objects interactively. According to [45], the engagement and enjoyment of students are high when in an AR-enhanced environment. The students tended to have a higher level of concentration when they dealt with the tasks.

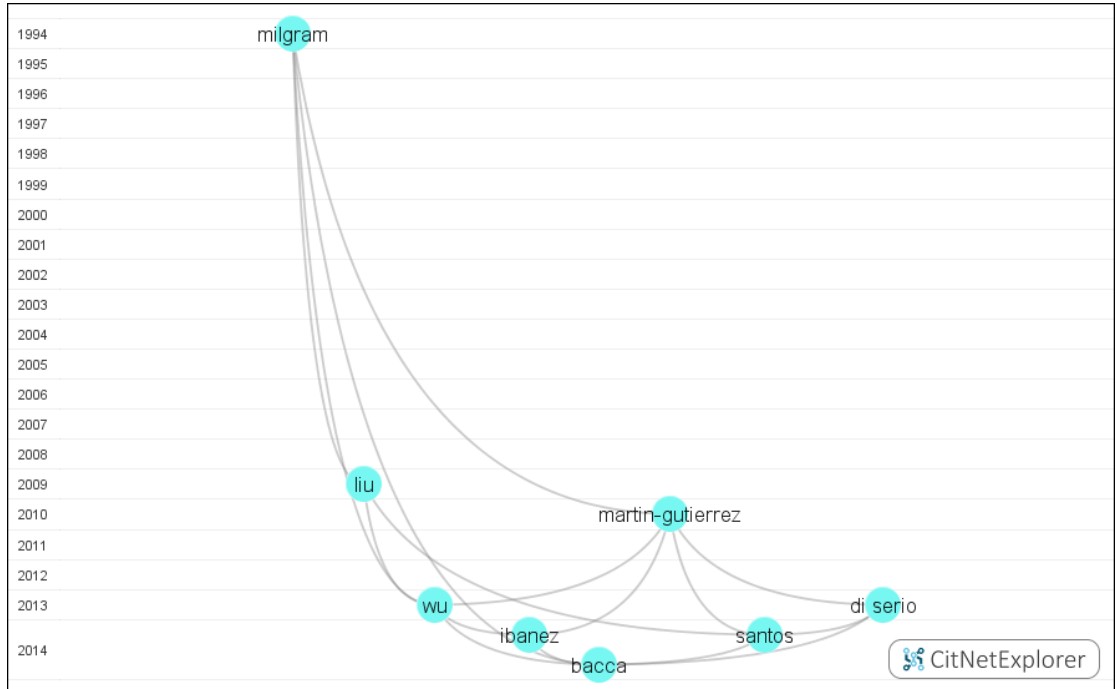

**Figure 4.** The visualization of the second-longest path analysis.

The main findings on the benefits of augmented reality are students' better learning outcomes [45]. Augmented reality technology can improve language skills and academic performance compared with traditional teaching methods. Learners who used information visualization showed a significant improvement in learning speed and quality and thus in their learning outcomes. According to [46], the use of AR technology in language teaching improves students' performance and makes vocabulary learning more effective. Moreover, the use of AR tools can also make a difference in the learner's emotions. On the one hand, it can improve students' positive emotions, such as satisfaction, enthusiasm, interaction with the environment and people, motivation, and positive perceptions of augmented reality use, making learning more effective [21,45]. On the other hand, it can also reduce learners' anxiety, increase their creativity, and foster their cooperation [39,47]. Therefore, it is noteworthy that the implementation of AR technology can have a significant positive impact on students' motivation to learn [48].

However, the use of AR tools can also cause some problems. Ref [37] found that if students are not offered well-designed interfaces, they will experience difficulty using the AR tools and understanding the information in the AR devices. Moreover, students will face the problem of cognitive overload because they have to deal with abundant information

and complex technology. In this way, students will become overwhelmed and stressed when they face these complex tasks. Ref [42] also noticed the constraints of using AR tools in that the current technology of AR is not enough to be applied in classes because of its insufficient memory, lower computing efficiency, and so on.

## 5. Discussion

RQ1 and RQ2 were devoted to the bibliometric analysis of the trend of AR tools in language learning. The annual trends of collected publications and citations can provide clear guidance for future research. RQ1 showed the trend of the research in the use of AR tools in language learning, which indicated its increasing popularity in the research field. Therefore, more research needs to be conducted in this area. The result of RQ2 showed the top ten authors, sources, organizations, and countries; it indicated that the development of AR tools in language learning is booming in the USA, Spain, Taiwan, and Turkey. Except for the USA and Spain, other countries, such as China, Australia, and Canada, are increasingly promoting their studies in AR learning. The study also figured outdetermined that the study of AR teaching and learning has led to more attention from researchers in Asian areas, and more studies from Asian organizations have promoted the development of AR learning in language learning. Additionally, the journal *Computers & Education* has contributed most to the development of AR learning. Therefore, researchers who are interested in AR teaching and learning would do well to analyze publications in *Computers & Education* for information on future research topics to fill gaps in previous research.

RQ3 was used to determine users' attitudes toward AR tools in language learning. Through the analysis of the first longest path, we found that both learners and teachers were in favor of using augmented reality in language learning. This could be attributed to the learner's self-perception of their improvement of language skills and their relief of negative emotions. This is consistent with previous studies. According to [47], the use of AR can help students reduce their anxiety levels, increase their creativity, and fosters collaboration and engagement in their real-life language environment. Therefore, learners tend to behave actively toward the use of augmented reality for learning English. A technology-enhanced approach was positively evaluated by students, as it enriched their language learning experience and at the same time increased their language awareness [49]. Thus, in future education, teachers should think about the method to develop students' positive attitudes toward modern teaching methods, which can be considered as a key factor for the integration of AR into language [35]. Moreover, educational institutions in the future should try to cultivate teachers' positive attitudes toward AR, which can appear to predict their perceptions of content creation and visualization using AR tools.

RQ4 explained the ways to integrate AR into language learning. The integration of AR tools in language learning is a key factor in the development of language skills. It was found that the most widely used augmented reality learning tool is HP Reveal, Vuforia, and Unity. In addtion, the well-integrated ways include 3D images, texts, games, and videos. A study found that these tools, when combined with games, can increase learners' motivation to learn [4]. One possible reason is that both games and AR tools focus on interaction, which plays a key role in language learning. For learners, they can use these language skills when interacting with the real world. The result may also be due to the rich scenes, voices, and characters in 3D images and videos, which make students more concentrated on the learning content itself and make the class more enjoyable to interact with. However, the limitation is that although the research on AR learning is continuously increasing, the actual use of AR tools in language has not been widely used in most schools. Therefore, the real effects of the integrated ways still need to be repeatedly verified.

QR5 investigated and proved the effect of AR on language learning. Augmented reality tools play an important role in language learning because they improve learners' performance, increase their motivation, and perceive the use of augmented reality positively. As a result of the AR-enhanced language learning, students' attitude toward learning English significantly improved and their self-efficacy in using English increased [48]. More-

over, the integration of AR tools in language learning contributes to the development of language knowledge and skills [39]. This result may be explained by the fact that immersive learning and increased motivation are the most important factors in language learning. By interacting with virtual information, learners gain a sense of immersion. As the use of AR allows for a deeper understanding of the learning material, learners can perceive things better in a context. In addition, learners feel relaxed and interested when interacting with AR tools. This stimulates their interest in learning and helps them increase their motivation to go deeper into the language. However, the use of AR tools can also cause some problems, such as cognitive overload on the students and the complexity of operation. This result has barely been described in previous research.

## 6. Conclusions

This part mainly includes the major findings, limitations, and implications for future research.

### 6.1. Major Findings

This study combined qualitative and quantitative research methods to analyze the use of AR in language learning. It summarized trends in the use of AR tools in language learning, users' attitudes toward AR tools, ways of integrating AR into language learning, and the effectiveness of the AR tools. The results of the study are based on longest path analysis. The yearly trends in the number of publications and citations showed increasing numbers in the field of AR in language learning, which indicated a bright prospect for the development of AR learning. This is consistent with the findings of [1] in that technology-supported language teaching will be offered more opportunities in future educational systems. At the same time, the top ten published authors, sources, organizations, and countries also indicated different research directions and priorities and provided an important point of reference for future researchers.

In terms of users' attitudes, both teachers and learners show positive attitudes toward the use of AR in language learning. Through the use of augmented reality tools, students showed more interest in the teaching and learning process [33]. Teachers' positive attitudes toward AR-supported teaching and learning influence their consistent use of AR in the classroom. Thus, their positive emotions make it easier to conduct the teaching activities and improve academic performance [24]. AR enhances language learning through 3D images, texts, videos, and games. The most widely used augmented reality learning tool is HP Reveal [4], an artistic platform that provides users with virtual content via their mobile devices. Research has shown that AR tools can facilitate language learning and improve students' performances by providing an immersive learning experience, increasing motivation, and reducing learning anxiety [4]. This finding agrees with the previous studies [21,23], which proved that the use of AR tools can have a positive effect on the learning process of students. However, there are also some disadvantages of the use of AR in language learning, such as cognitive overload and the complexity of operation [37]. Therefore, teachers need to pay attention to the use of AR in the teaching process.

### 6.2. Limitations

Due to the authors' limited knowledge, there are some limitations. First, because this study did not register for a protocol, there may be cognitive bias in the process of the analysis. Second, the study may not contain all related publications because CitNetExplorer can only analyze the resources from the Web of Science. Third, this study only analyzed positive attitudes toward the use of AR; there may also be a few studies about negative emotions toward AR tools. Moreover, there may be other ways to integrate AR tools with language learning. However, this study only introduced the three most common approaches, which are 3D images, videos, and games. More advanced ways should be further explored in the future. Lastly, this study mainly analyzed the effect of AR on language learning for students. With the help of AR tools, students can improve their

language skills and knowledge. However, whether or not the use of AR in language teaching can exert an influence on teachers was not involved in this study.

*6.3. Implications for Future Research*

The findings indicate that a positive attitude plays an important role in AR-enhanced language learning. Therefore, in the future educational field, teachers should cultivate students' interest of learning language through the use of AR tools. In turn, students can improve their performance in language learning with the help of AR tools. Additionally, the findings also show that AR enhances language learning through the use of 3D images, texts, videos, and games. Thus, when future teachers conduct their vocabulary teaching with AR tools, they can present the words using 3D images in order to deepen students' understanding and memorization of these words. According to this study, the lack of real context affects students' language learning. At this point, AR can solve this problem by helping learners see virtual objects that are composited within the real environment. Therefore, to foster students' cognitive development of language learning, the future educational system can apply teaching activities that are supported by AR in the language teaching process.

Future research should realize the importance of educators' and practitioners' attitudes. This study found that both educators and students have positive attitudes toward the use of AR in language learning. Therefore, in terms of educators and educational institutions, the factors that contribute to the formation of positive emotions should have more attention paid to them. Furthermore, future studies should explore whether negative emotions exist with respect to the use of AR tools. The reasons for the negative emotions should also be explored, which will make sense for future language education. Both teachers and students should notice that their attitude toward AR tools will directly affect their perception and mastery of language skills. Additionally, further educational practices could explore the attitude of different age groups and different educational levels based on this study.

With the technological advancement in AR, more and more AR applications will be invented, and integration with AR tools will become a major approach. Thus, future studies can explore the effectiveness of various AR applications. Researchers can think about more approaches for different educational levels to integrate new versions of AR tools for language learning. In the future, researchers could explore the effect of the use of AR tools in language learning from the perspective of educators. According to this study, educators' attitudes toward AR-enabled instruction affect their continuous use in the classroom. Therefore, future studies can collect teachers' feelings and feedback about the AR tool, upon which more effective AR applications can be invented.

**Author Contributions:** Conceptualization, Z.Y.; Methodology, W.M.; Investigation, W.M.; Writing—original draft, W.M.; Writing—review & editing, W.M.; Funding acquisition, Z.Y. All authors have read and agreed to the published version of the manuscript.

**Funding:** This work is supported by 2019 MOOC of Beijing Language and Culture University (MOOC201902) (Important) "Introduction to Linguistics"; "Introduction to Linguistics" of online and offline mixed courses in Beijing Language and Culture University in 2020; Special fund of Beijing Co-construction Project-Research and reform of the "Undergraduate Teaching Reform and Innovation Project" of Beijing higher education in 2020-innovative "multilingual +" excellent talent training system (202010032003); The research project of Graduate Students of Beijing Language and Culture University "Xi Jinping: The Governance of China" (SJTS202108).

**Institutional Review Board Statement:** Not applicable.

**Informed Consent Statement:** Not applicable.

**Data Availability Statement:** Not applicable.

**Conflicts of Interest:** The authors declare no conflict of interest.

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
