# Peer review of "A Bibliometric Analysis of Augmented Reality in Language Learning"

_sustainability, doi:10.3390/su15097235_

Round 1

Reviewer 1 Report (Previous Reviewer 1)

The revised manuscript has undergone significant improvements in its quality. From my evaluation, I see potential for the article to be accepted for publication in Sustainability. In light of this, I recommend that the authors augment the manuscript with additional implications and suggestions for future educational practices based on the study's findings.

Author Response

Thanks for your comments and suggestions. Please see the attachment.

Reviewer 2 Report (Previous Reviewer 2)

Point 1.

Authors:

Before the response to the comments, we want to apologize for our careless neglect of the first-version comments.  

For the first suggestion, we have searched the registration criteria.

According to the protocol inclusion criteria in PRISMA,

the systematic review needs to have health related outcomes.

The current study is mainly about language learning. Therefore, this study don’t have to register for a protocol.

Answer:

No, this is not true. The PRISMA statement (2009) states that

"some modifications of the checklist items or flow diagram will be necessary in particular circumstances."

There is multiple exemple of the PRISMA statement applied to other domain than health care (e.g., 

Immersive Virtual Reality for Foreign Language Education: A PRISMA Systematic Review).

The authors misunderstood my point. There is nothing wrong doing a systematic review which is not registered, 

but it does not have the same level of proof that a registered one. In particular, registered review avoid the cognitive bias, 

consisting of changing the scope and methodology of the review during research to match or alter the results. 

In order to inform the reader, who can be mistaken, the authors should simply add a statement in their discussion/conclusion to explain this limitation.

Point 2:

Authors:

Thanks for your comments and suggestions.

We have added more details of the cluster tools and the scope of this study.

The number and size of the cluster have been presented in Table.

Answer:

No, the authors misunderstood my point.

The clustering algorithm from Van Eke et al. accepts a resolution parameter, and a minimal cluster size.

It is common habit in ML field to report and justify the choice of the hyperparameters. 

While they are at it, the authors should also mentioned the direction problem (i.e., "no distinction is made between publication i citing publication j and publication j citing publication i", van Eke)

Point 3:

Other comments on confusing definitions of AR/VR.

Answer:

I acknowledge the effort made by the authors to better explain augmented reality. 

This considerably improved the reading, as previously the definition provided in the introduction was different from the scope of the query.

At some locations however, the old minding still appears, eg. :

- "In addition, VR videos have also been used to foster language learning.

Chen (2020) conducted a study using AR videos to support students’ English language

learning."

- Bahari 2022, is a reference on virtual reality not augmented reality.

In the continuity of Milgran and Kishino's reality continuum, I recommend the authors to deepen their introduction, 

and their understanding of the reality continuum. It is not only necessary to provide a definition for augmented reality, but also

to briefly explain where it fits in the AR/VR/MR/Modulated and retransmitted realities world (see e.g. Virtual Reality: Definition and Craze, 2021).

This will add more clarity to your article.

Author Response

Thanks for your comments and suggestions. Please see the attachment.

Reviewer 3 Report (Previous Reviewer 3)

After study revised version i recommend accept this article in current form.

Author Response

Thanks for your comments and suggestions. Please see the attachment.

Reviewer 4 Report (New Reviewer)

First, I appreciate the authors for developing this study on this topic concerning augmented reality and language learning. However, this manuscript needs some careful modifications. Specifically, I have the following remarks. 

- The abstract section needs to be catchy. 

- Source reference for Table 1 should be mentioned. 

- Please add the structure/organization of this study in the last paragraph of the introduction section. 

- Please make corrections in the sentence, "QR5 aims to explore the effect of AR on language learning." [Line No. 461] 

- It's not a very good practice to put findings inside the conclusion section. Please place the findings before the conclusion. 

- Why did you choose the Web of Science? Why not ScienceDirect/IEEE Xplore/Scopus? Please provide justifications. 

- Corroboration has not been seen in the conclusion section. So, please include corroboration to further strengthen your results. 

- Lastly, the author(s) should review and cite the following very related and relevant paper that has been recently published in Sustainability. 

Paper link: https://www.mdpi.com/2071-1050/15/2/1470 
Paper title: Blockchain in Online Learning: A Systematic Review and Bibliographic Visualization 

Author Response

Thanks for your comments and suggestions. Please see the attachment.

Round 2

Reviewer 2 Report (Previous Reviewer 2)

Please, move the following section from Results to Method (under 3.2): 

"The clustering parameters contain a resolution parameter, which is 1.00, and the minimum cluster size, which is 10. Besides, the optimization parameters include the number of random starts, which is 1, the number of iterations, which is 10, and the random seed, which is 0. Due to the defined minimum cluster size, 94 publications were not included in the clustering."

Author Response

Thanks for your comments and suggestions. Please see the attachment.

This manuscript is a resubmission of an earlier submission. The following is a list of the peer review reports and author responses from that submission.

Round 1

Reviewer 1 Report

Many thanks for submitting your manuscript to Sustainability. Here and now, the study has significance and appropriateness. The study has the potential to make contributions to the field and knowledge. However, to be considered and published in a SSCI journal, some issues need to be addressed very carefully as follows:

The significance of the study needs to be articulated explicitly, especially for potential international readers.

The contributions of the study (methodological, theoretical, and practical contributions) need to be articulated explicitly.

The discussion needs to be more comprehensive to engage the study in a meaningful conversation with past scholarly works for outshining the contributions of the study to the knowledge and field.

Why do the researchers use the Bibliometric Analysis approach? What is the advantage of the method? The researchers need to justify the reasons for choosing this particular method over other approaches.

There are significant gaps between research questions. To some extent, RQ1 and RQ2 are closely related. However, RQ3, RQ4 and RQ5 need to be linked to the previous two RQs.

The results and discussion sections of RQ3, RQ4, and RQ5 are far from sufficient to make solid and meaningful arguments. I would suggest that the authors expanding the sections.   

I would suggest the authors make some practical suggestions for future educational practices, such as for educational institutions, educators, for practitioners of different educational levels, and from different cultures/countries.

Reviewer 2 Report

The authors intend to draw research directions for the use of augmented reality in language learning, based on bibliometric analysis.

The authors' approach is interesting and original. However, I see some issues with the current version of the manuscript:

1) "Games, 3D images, videos are the main ways to immerse learner[s] in a virtual world, which play an important role in language learning" (abstract)

-> This is, in fact, quite the opposite: AR is about instancating virtual elements into the real word (the definition provided is a better match for VR).

2) The reader could benefit from a small paragraph explaining in a few words the advantages of using AR (or mediated reality in a broader perspective) to learning: e.g., better stimulation of visual memory, embodiment...

3) Legend of figure 1 should be in plain English

4) The authors should provide some insight into the clustering method inside CiteNetExplorer and, more importantly, its limitation.

5) The study is apparent to a systematic review, but the protocol was not registered. 

6) The scope of the query seems quite limited. I know for sure, that researchers in Netherlands was working on this (e.g., Tilo Hartmann, Angelica Tinga, ...). 

Reviewer 3 Report

Presented article with Title ”A Bibliometric Analysis of Augmented Reality in Language 2 Learning” is writing on 13 pages with 4 figures, 2 tables and 63 references. The paper is written clearly, but it has a number of shortcomings. The structure is very clear (introduction, Literature review, methods, results, discussion and conclusion).

Suggestions:

-          In article is missing a more detailed art of state.

-          In the article needs to reformat the links to the sources according to the journal template.

-          I recommend the authors to pay more attention to methodology.

-          Figure 1 please translate legend of figure.

-          Please more expain figures (2,3,4).

-          What is the benefit of the presented article?    

-          Conclusions require redrafting. They are too short and inconsistent.

The quality and information provided in some figures needs correcting. All the specific comments can be followed in revised copy of the manuscript.     

I recomend this paper publish in journal after minor revision.

Round 2

Reviewer 1 Report

The quality of the manuscript has been significantly improved based on the reviewers’ comments.
However, the authors are encouraged to state clearly about the contributions of the current study, including methodological contributions, theoretical contributions, and practical contributions to the field and knowledge.

Author Response

(The authors gave the same response as above.)

Reviewer 2 Report

Let's put apart the fact that the authors ignore one of my comments about this study not being registered before the analysis (which is, by the way, a guarantee that biases are not introduced, by changing the query AFTER analysis): The authors could have just kicked the ball into touch by stating the limitation, instead of ignoring a valid comment.

Most importantly, I don't find the authors' answers convincing. I don't see that the authors have a good understanding of the clustering algorithm they used or this limitation (what about the choice of the hyperparameter of their model, such as the number and size of the clusters?). 

Finally, the authors chose to answer my doubt about the query scope by stating that not all articles are in Web of Science. I am going to reply that it is instead the authors' query which is not very well defined, starting by their definition of "augmented reality" which is 1) never defined properly in the paper and 2) confused with the notion of virtual reality, (e.g. "AR enhances language learning by immersing learners in a virtual world with 3D images, movies, and games"). The authors reference Milgran and Kishino's reality continuum but actually never read the paper.

To put it into a nutshell, I see some potential in this paper and an original approach. But I am not at all convinced by the author's answer.

Author Response

(The authors gave the same response as above.)
